# Electroretinogram Analysis Using a Short-Time Fourier Transform and Machine Learning Techniques

**DOI:** 10.3390/bioengineering11090866

**Published:** 2024-08-26

**Authors:** Faisal Albasu, Mikhail Kulyabin, Aleksei Zhdanov, Anton Dolganov, Mikhail Ronkin, Vasilii Borisov, Leonid Dorosinsky, Paul A. Constable, Mohammed A. Al-masni, Andreas Maier

**Affiliations:** 1Engineering School of Information Technologies, Telecommunications and Control Systems, Ural Federal University Named after the First President of Russia B. N. Yeltsin, 620002 Yekaterinburg, Russia; a.e.zhdanov@urfu.ru (A.Z.); anton.dolganov@urfu.ru (A.D.); m.v.ronkin@urfu.ru (M.R.); v.i.borisov@urfu.ru (V.B.); l.g.dorosinskiy@urfu.ru (L.D.); 2Department of Artificial Intelligence and Data Science, College of Software & Convergence Technology, Daeyang AI Center, Sejong University, Seoul 05006, Republic of Korea; 3Pattern Recognition Lab, Department of Computer Science, Friedrich-Alexander-Universität Erlangen-Nürnberg, 91058 Erlangen, Germany; andreas.maier@fau.de; 4College of Nursing and Health Sciences, Caring Futures Institute, Flinders University, Adelaide, SA 5042, Australia; paul.constable@flinders.edu.au

**Keywords:** electroretinography, biomedical signal processing algorithms, short-time Fourier transform, spectrogram, feature extraction, classification, machine learning, deep learning, neural network, retinal study

## Abstract

Electroretinography (ERG) is a non-invasive method of assessing retinal function by recording the retina’s response to a brief flash of light. This study focused on optimizing the ERG waveform signal classification by utilizing Short-Time Fourier Transform (STFT) spectrogram preprocessing with a machine learning (ML) decision system. Several window functions of different sizes and window overlaps were compared to enhance feature extraction concerning specific ML algorithms. The obtained spectrograms were employed to train deep learning models alongside manual feature extraction for more classical ML models. Our findings demonstrated the superiority of utilizing the Visual Transformer architecture with a Hamming window function, showcasing its advantage in ERG signal classification. Also, as a result, we recommend the RF algorithm for scenarios necessitating manual feature extraction, particularly with the Boxcar (rectangular) or Bartlett window functions. By elucidating the optimal methodologies for feature extraction and classification, this study contributes to advancing the diagnostic capabilities of ERG analysis in clinical settings.

## 1. Introduction

Electroretinography (ERG) is a non-invasive form of assessing the functional health of the retina through its response to light stimulation. The stimulation is presented as a series of interval-based light pulses, which trigger varying responses based on the state of retinal adaptation and the wavelength, duration, strength, and stimulating frequency of the light pulse [1]. Cone photoreceptors, which are responsible for photopic or ’daytime’ vision, require more quanta for activation compared to the rod photoreceptors that function in scotopic or ’night-time’ vision, requiring fewer quanta to activate [1,2]. The responses from the photoreceptors and post-receptoral neurons (bipolar, horizontal, amacrine, and ganglion cells) all contribute to the overall size and shape of the recorded one-dimensional (1D) ERG signal [1,3].

Several ERG recording methods, including full-field flash, pattern, and multifocal, can be utilized for the early detection and diagnosis of a wide variety of retinal-related diseases, including early diabetic retinopathy, glaucoma, retinal dystrophies, and age-related macular degeneration [1,2,4,5,6].

Typically, the full-field ERG (ffERG) signals last about 250 ms and have a frequency range of 0 to 300 Hz [7]. This test is crucial for assessing the functionality of the retina, which is essential for vision. As illustrated in Figure 1, the ERG signal primarily consists of two main components: the a-wave and the b-wave. The a-wave is the initial negative deflection in the ERG signal and is generated by the retina’s photoreceptor cells (rods and cones). Following the a-wave, the b-wave is a positive deflection produced by the inner retinal cells, mainly the bipolar and Müller glial cells. These waves are crucial for understanding the retina’s response to light stimuli. The main characteristics of these waves are their amplitudes and time to peak. The amplitude of the a-wave (Va) and the b-wave (Vb) refers to the height of these waves, measured in microvolts (μV). These amplitudes reflect the strength of the response generated by the retinal cells. The time to peak of the a-wave (Ta) and the b-wave (Tb) represents the time it takes for these waves to reach their maximum height after the light stimulus, measured in milliseconds (ms) [1]. These time-domain features are essential for diagnosing various retinal conditions. In addition to the main a- and b-wave components, the ERG signal can also include other components such as the Oscillatory Potentials (OPs) and the Photopic Negative Response (PhNR). The OPs are high-frequency wavelets superimposed on the ascending limb of the b-wave. They are thought to originate from the inner retinal layers, particularly the amacrine cells, and are useful for evaluating inner retinal function. The PhNR is the negative wave following the b-wave peak and is shaped by the retinal ganglion cells. An additional test protocol is the Flicker ERG, recorded using a pulse presented at 30 Hz. The response primarily assesses the cone system’s functionality; it is responsible for color vision and visual acuity under photopic conditions and is useful for diagnosing cone-related disorders [8].

A number of different ERG signals may be extracted based on the different electrophysiological protocols and clinical applications [9]. The scotopic 0.01 ERG response is obtained under dark-adapted conditions and is generated by rod photoreceptors with a dominant b-wave and minimal a-wave. The scotopic 2.0 ERG has a stronger flash strength when presented under dark-adapted (DA) conditions and has a mixed rod–cone response. Under light-adapted (LA) conditions, the photopic 2.0 ERG response is a cone-driven response of typically smaller amplitude owing to the fewer cones in the human retina [1].

Currently, the most widely used form of ERG analysis and feature extraction is time-domain analysis, which involves the identification of the *a*- and *b*-wave amplitudes and their corresponding time to peaks, usually by algorithms that find the peaks automatically that can then be checked by the clinician [1,2]. However, the time-domain features do not fully reveal the underlying energy contributions of the neural generators (photoreceptors, bipolar, amacrine, horizontal, and retinal ganglion cells). So, alternative methods using signal analysis have been explored to deconstruct the signal further [10]. These methods include Power Spectral Density (PSD) and Fourier Transform, as well as time–frequency-domain methods such as the Short-Time Fourier Transform (STFT), and Continuous and Discrete Wavelet Transforms [2]. Although these methods have yet to be explored as extensively as the time domain, they offer a more detailed analysis and additional features than those provided by pure time-domain analysis.

Regarding time–frequency analysis, the predominant research has been on Wavelet Transforms, with limited exploration of the Short-Time Fourier Transform (STFT) in analyzing ERG signals. Thus, this study uses STFT as an additional signal analytical approach to the ERG. As referenced in Section 2, the existing literature has predominantly employed STFT as a complementary technique to Wavelet Transform methods as a comparison. This highlights the opportunity to delve deeper into the potential benefits and insights that STFT could offer in the overall analysis of ERG signals.

STFT could be selected as the most interpretable of the transformations mentioned above. The spectrogram is a 2D representation of the signal with the time on the horizontal axis and the frequency on the vertical axis, which can be given as follows:(1)STFT(τ,f)=FFT(x(t)·w(t−τ)),
where FFT is the fast Fourier Transform of spectrum calculation; STFT(τ,f) corresponds to the representation of the input signal *x* with the window function *w* (with given length and form) for time position τ and frequency position *f*. Let us denote that in general the spectrum can be expressed as $\int_{-\infty }^{+\infty }x\left(t\right)w\left(t-\tau \right)e^{-j2\pi ft}dt$ [11].

Equation (Equation 1) provides a linear, unambiguous, and reversible relationship between the input (*x*) and output results (STFT(τ,f)). The power spectrum density for the next processing can be given as |STFT(τ,f)| due to the complex origin of the equation [11].

STFT uses a sliding overlapping window function to convert the signals to a time–frequency domain using the fast Fourier Transform (FFT) algorithm. This produces a 2D spectrogram representation with the time on the horizontal axis, the frequency on the vertical axis, and the amplitude/power represented as a color map. Figure 1 depicts a healthy (top) and an unhealthy (bottom) signal with dystrophy in the time domain and their corresponding spectrogram representations calculated using STFT. We can see that the spectrogram gives us the signal frequency, which is in the range of 0–100 Hz, along with the time the frequency occurs and how much power that frequency contains, with red representing higher power frequencies and blue representing lower power frequencies.

The STFT spectrogram shows the energies within each frequency band from 0 to 100 Hz. The horizontal axis of the spectrogram denotes time bins (in milliseconds), and the vertical axis represents frequency bands in Hertz (Hz). It should be noted that the black arrows in Figure 1b show that the spectrogram frequency distribution is from 0 to 50 ms and 0 to 20 Hz (maximum energy), from 60 to 80 ms and 0 to 10 Hz (medium energy), and from 0 to 67 ms and 15 to 30 Hz (low energy). The black arrows in Figure 1d show that the spectrogram frequency distribution is from 0 to 80 ms and 0 to 15 Hz (maximum energy), from 0 to 80 ms and 15 to 25 Hz (medium energy), and from 25 Hz (low energy). The key difference between the healthy signal (Figure 1b) and the unhealthy signal (Figure 1d) is evident in the energy distribution across the frequency bands. The healthy signal shows a more diverse energy spread with the maximum energy occurring at higher frequencies (0–20 Hz) compared to the unhealthy signal, where the maximum energy is concentrated at lower frequencies (0–15 Hz). This indicates that unhealthy signals tend to have more energy concentrated in the lower frequency bands, suggesting a potential marker for identifying signal health.

This study compared various window functions for optimal feature extraction using STFT and spectrogram generation to classify the signals and determine which window yielded the best features for ERG signal classification. Several combinations of window function, window size, and window overlap were used to extract spectrogram images to train deep learning (DL) models, and manual feature extraction, which was used to train classical machine learning (ML) models. The results from both approaches were compared to determine which window yielded the best signal classification and whether DL had an advantage over the classical ML approaches. The main contributions of this study were the use of different window parameter combinations for feature extraction and the application of DL for classifying the extracted spectrogram images.

The paper is organized as follows: Section 2 reviews relevant studies and those employing STFT for feature extraction in similar fields. Section 3 presents the materials and methods used for the study, which include the ERG signal database and the pipeline for signal processing, feature extraction using the STFT, model building, and evaluation. Section 4 and Section 5 describe the results obtained from the analyses using multiple evaluation metrics and discuss the outcomes, and, finally, Section 6 concludes with the implications of the findings for the analysis of the ERG and future directions.

## 2. Related Works

ERG analysis methods can be divided broadly into three different approaches: time-domain analysis, which involves analyzing the amplitudes and time to peaks of the signal; frequency-domain analysis, which involves studying the frequencies of the signal; and time-frequency analysis, which involves studying the signal’s frequencies at the time they occur along with their power and nonlinear methods [2]. Time-domain analysis, for the most part, is the most popular method used in the literature because it is fast and usually provides differences in amplitude or time to peak when there is a retinal disease. However, subtle or early functional changes may not be evident in time-domain analysis initially, such as in diabetes and glaucoma, thus the application of signal analysis may improve earlier diagnosis in both. In addition, signal analysis may also support classification between groups in early neurological disorders [10].

Several studies have used frequency-domain methods to analyze ERG signals. These methods provide a different perspective on signal analysis by providing spectral information unavailable in the time domain. Most studies in the frequency domain use the FT with the FFT algorithm [12] to convert the signal into the frequency domain before analyzing it. A few other methods, namely Power Spectral Density (PSD) and Linear Prediction (LP), have also been used. In [7,13], Gur et al. were able to find similarities between corneal and non-corneal ERG signals by using FFT and LP to identify specific frequencies in normal corneal ERGs under different conditions. After studying the Oscillatory Potentials (OPs) from the ERG signals of diabetic patients using the FFT, Vander Torren et al. [14] concluded that it was possible to express OPs quantitatively even in pathologies. Similarly, by studying photopic and scotopic ERGs in the Fourier spectrum and comparing them to the time domain, Li et al. [15] were also able to highlight differences in the dominant frequency and power between the scotopic and photopic ERGs. In a different study, Sieving et al. [16] used discrete Fourier Transform (DFT) to study Flicker ERGs cycle by cycle, extracting real-time harmonic components.

Using Welch’s Power Spectral Density (PSD), Karimi et al. [17] were able to find significant differences in the frequency components in the scotopic and photopic ERGs of patients with and without retinitis pigmentosa. To search for signs of retinal pathologies in patients with stage I and II open-angle glaucoma, Zueva et al. [18] analyzed the frequency responses from Flicker and pattern ERGs by decomposing them into a Fourier Series.

While the frequency-domain methods mentioned above provide spectral information about ERG signals, they need to improve significantly regarding temporal information, which is crucial for ERG analysis. Time - frequency domain methods offer a way to obtain both spectral and temporal information from the signals and represent it in a 2D or 3D format. Unlike the classical FT, STFT allows us to visualize the signal’s frequencies, the time window at which they occur, and how strong that frequency is at that point in time. This allows us to extract multi-dimensional features that are otherwise not accessible in the time domain or frequency domain alone.

To the best of our knowledge and findings, virtually all time-frequency ERG analysis studies are based on Continuous and Discrete Wavelet Transforms, except very few studies that included STFT as part of the analysis. In [19], STFT was one of the time–frequency methods used along with Continuous Wavelet Transform (CWT) and Discrete Wavelet Transform (DWT) to analyze the photopic ERG signals obtained from a healthy subject. In [20], STFT was applied along with CWT and DWT to analyze the effects of obesity on ERG signals. Three different responses (cone, rod, and maximal combined) were analyzed, and features were extracted using STFT, CWT, and DWT, after which the results from these methods were compared. In [21], STFT and DWT were used to determine the frequency components of the three photopic and Flicker 30Hz ERG signals of patients with Central Retinal Vein Occlusion (CRVO). More recently, ref. [8] used CWT to manually extract features from adult and pediatric signals, which were used to train a Decision Tree classifier by combining time-domain features (a- and b-wave amplitudes and implicit times) with the wavelet features. In a similar study to this paper, ref. [22] compared several mother wavelet combinations to determine which combination would better classify pediatric ERG signals.

It is worth noting that a significant drawback of STFT is its time–frequency resolution trade-off, which stems from the uncertainty principle (Gabor limit in signal processing) [23]. This means that it is impossible to achieve a high resolution for both the time and frequency components of the signal simultaneously; hence, a compromise has to be made between the two. Thus, the larger the window size, the better the frequency resolution and the lower the time resolution, and the smaller the window size, the better the time resolution, but the lower the frequency resolution.

Table 1 provides several frequency and time–frequency domain methods used for analyzing ERG signals previously, as well as the types of ERG signals used for the studies.

The analyses presented in Table 1 indicate a prevalent preference among researchers for the FT in frequency-domain analysis and the Wavelet Transform in time–frequency-domain analysis. This inclination towards the Wavelet Transform in time–frequency-domain analysis may stem from considering the time–frequency resolution trade-off. As different mother wavelets in Wavelet Analysis can impact signals uniquely, the STFT also exhibits variations in signal representation based on factors such as the chosen window function, window size, and overlaps between windows during signal processing. This warrants further investigation to ascertain the efficacy of STFT in ERG signal analysis, given its nuanced response to different signal characteristics.

## 3. Materials and Methods

### 3.1. Dataset

The database used for the study consisted of five ERG signal types: Maximum 2.0 ERG response, Photopic 2.0 ERG response, Scotopic 2.0 ERG response, Photopic 2.0 ERG Flicker response, and Scotopic 2.0 ERG Oscillatory Potentials. The database consisted of pediatric and adult patients with the signals recorded according to the ISCEV recording standard [1]. For detailed descriptions of the database and protocols employed in this study, please refer to [24]. All recordings in the dataset were made with the Tomey GmbH EP-1000 stimulator sampling at 2 kHz with a 0.1–300 Hz bandpass filter. The DTL fiber active electrode was utilized for ERG recordings. The DTL was composed of 7 cm long, low-mass spun nylon fibers impregnated with metallic silver. However, despite its comfort, DTL electrodes are commonly used in ERG due to their poor stability on the eye, leading to potential movement with blinks and subsequent variations in amplitude. Conversely, gold foil or contact lenses offer greater stability but are less comfortable. Thus, each type of electrode in ERG exhibits distinct strengths and weaknesses [25]. Two flash strengths were employed: DA 2.0 for scotopic responses and LA 2.0 for maximal responses. Flash stimuli consisted of a white light at 2 cd·s·m^−2^ intensity on a 0 cd·s·m^−2^ background, ensuring standardized conditions for eliciting retinal responses [26].

This study utilized only the scotopic maximum (DA2) ERG response signals from the dataset. DA2 has a duration of up to 250 ms. However, most samples in the dataset used for this study have a length of 100 ms, with some extending up to 250 ms. To maintain consistency and reduce noise, we truncated the longer signals to match the size of the majority of samples. The signals in the dataset provide information on diseases such as cone and rod retinal dystrophies. More details about the dataset are shown in [24].

Due to the highly unbalanced nature of the dataset, with the unhealthy samples being more than twice the number of the healthy samples, we used a balanced version of the dataset to avoid having a biased and overfitted classifier. The dataset balancing was accomplished using the Imbalanced Learn Python library [22]. In this study, we utilized an under-sampling technique with the A11KNN function from the library, which uses the nearest neighbor algorithm to identify and contradict samples and their neighborhoods [27]. Table 2 demonstrates the distribution between the healthy and unhealthy signals in both the unbalanced and balanced datasets.

### 3.2. Spectrogram Conversion

Figure 2 shows the stages undertaken for the entire methodology. The major stages of the pipeline were the image and feature extraction stages, the classical ML stage, and the final DL stage.

The first stage involved data preparation before spectrogram image extraction for the DL models. The ERG signals were split into healthy and unhealthy classes. We then employed an 80:20 split for training and testing, with a training set further divided into 90:10 training and validation sets, resulting in a final 70:20:10 split of training, testing, and validation, respectively. This splitting occurred on the raw data rather than the extracted images to prevent data leakage, ensuring all images from the same signal remained together in either the training, testing, or validation sets. We then utilized various window functions to extract spectrograms, namely Boxcar, Hann, Hamming, Tukey, Bartlett, Blackman, Blackman–Harris, and Taylor windows [28]. The entire process, including data splitting and spectrogram extraction, was implemented using Python (version 3.11). Finally, the images were organized according to the Imagenet dataset structure [29].

#### 3.2.1. Image Extraction

For the DL classifiers, spectrogram images were calculated using the STFT. Multiple spectrogram images were extracted for each signal using different combinations of the window function, size, and overlap to obtain a large number of images for training the DL models. Given that each window has 21 size–overlap combinations and a total of 120 signals, this yielded 2520 spectrogram images for each window. Each window contained several sizes and overlaps used to extract the spectrogram images. The window sizes and overlaps were chosen as powers of 2, as recommended by the FFT algorithm [30]. This was implemented due to the time–frequency trade-off inherent in the conversion of signals, whereby signals with larger window sizes would yield better frequency-domain resolution (i.e., better feature representation), while signals with smaller window sizes would yield better time-domain resolution.

As previously described, the data was split prior to image extraction to avoid data leakage. All images from a single signal were saved in one location; hence, if one signal was in the validation set, then all images from that signal were in the same validation set.

#### 3.2.2. Feature Extraction

The obtained spectrogram arrays were processed to extract specific features: maximum, minimum, median, and mean intensity values (bmax, bmin, bmedian, and bmean, respectively). This extraction process is essential for a thorough analysis of the ERG signals. Spectrograms are visual representations of the spectrum of frequencies in a signal as it varies with time. By converting ERG signals into spectrograms, we can analyze the frequency components and their changes over time, providing a deeper understanding of the retinal responses. The maximum intensity value (bmax) represents the highest energy point in the spectrogram, which can indicate the most robust response at a particular frequency. The minimum intensity value (bmin) shows the lowest energy point, helping to identify the baseline activity or noise level. The median intensity value (bmedian) provides the middle value of the intensity distribution, offering a measure of the central tendency that is less affected by outliers. The mean intensity value (bmean) gives the average energy level across the spectrogram, offering insight into the overall energy distribution of the signal. These features (bmax, bmin, bmedian, and bmean) are crucial for various reasons. First, they help quantify the ERG signal’s characteristics, making comparing signals from different patients or conditions easier. Second, these features can be used as input for ML algorithms to classify or predict retinal conditions based on ERG data. Finally, by analyzing these intensity values, researchers can identify patterns or abnormalities that might not be apparent in the time-domain analysis alone.

The features for each spectrogram output were extracted separately so that each window-size–overlap combination was distinct from the others. Table 3 shows the window sizes and overlaps used for the signal conversion. With 21 size–overlap combinations for each window, there were a total of 168 parameter combinations, from which the features of each resulting spectrogram were individually extracted. Figure 3 shows a spectrogram representation of signals based on windows used for the study with a window size of 32 and an overlap of 16. Each window represented the signal in a slightly different form based on the shape of the window used, namely (a) Hamming, (b) Hann, (c) Boxcar, (d) Bartlett, (e) Blackman, (f) Blackman–Harris, (g) Tukey, (h) Taylor.

### 3.3. Machine Learning Classifiers

Figure 2 demonstrates the training pipeline used in both the ML and DL approaches. As mentioned in Section 3.2.2, the ERG signal was first converted into a spectrogram from which the features bmin, bmax, bmedian, and bmean were extracted. This was obtained for each combination of window-size–overlap parameters. At each iteration, a different parameter combination was used to obtain the spectrogram before extracting the features for the classifiers.

Two classifiers were used for the standard ML approach: Decision Tree (DT) and Random Forest (RF) algorithms. These classifiers were implemented using the Scikit-Learn ML library [31]. These classifiers were chosen due to their ability to prioritize the most significant features in the feature set for the task. In addition to the DT classifier being able to prioritize the most significant features, the RF classifier, being an ensemble of multiple DT classifiers, can take advantage of various DTs, capitalizing on their advantages while minimizing their disadvantages. Table 4 shows the parameters used for the classifiers.

We applied a 5-fold stratified k-fold cross-validation for the model training and tuned the hyperparameters to observe whether the classifier could yield better results.

### 3.4. Deep Learning Classification Models

ML techniques often struggle with the multifaceted nature of spectrogram-transformed data. ML methods require extensive feature engineering to identify relevant features, which is a time-consuming process and prone to human selection bias [32]. In contrast, DL offers a promising alternative, leveraging its ability to learn hierarchical representations automatically [33]. In this study, we used classical architectures: DenseNet121 [34], ResNet50 [35], VGG16, and VGG19 [36], as well as a new robust architecture, Visual Transformer (ViT) [37], that has been used for ERG classification in the time–frequency domain [22,38]. For this analysis, we use ViT Small (ViT_small_r26_s32_224) [39]. ViT is available at the HuggingFace Transformers repository [40]. The remaining models are available at the HuggingFace Timm repository [41].

We used an ADAM [42] optimizer with an initial learning rate of 0.001 for all the models. Each model was trained until convergence using early stopping criteria based on the validation loss. The cross-entropy loss function was utilized for the training. All experiments were performed on a single NVIDIA A100 graphics processing unit on a machine with two Intel Xeon Gold 6134 3.2 GHz and 96 GB RAM.

Random crop, translation, rotation, horizontal flip, and vertical flip were exclusively utilized on the images under consideration for the augmentation [43].

### 3.5. Metrics

To analyze the performance of the models, several metrics, including Accuracy, Precision, Recall, and F1-score, were calculated. These metrics provide a comprehensive view of the performance of each model:(2)$$Accuracy=\frac{TP+TN}{TP+FP+TN+FN},$$
(3)$$Precision=\frac{TP}{TP+FP},$$
(4)$$Recall=\frac{TP}{TP+FN},$$
(5)$$F1\;Score=\frac{2\times Precision\times Recall}{Precision+Recall},$$
where TP=TruePositive, TN=TrueNegative, FP=FalsePositive, FN=FalseNegative. We also evaluate the performance of each classification model using the receiver operating characteristic (ROC) with its area under the curve (AUC).

Combining these metrics in a binary diagnostic classification problem ensured a comprehensive model performance evaluation. Accuracy offers an overall success rate, where Precision and Recall become critical by focusing on the model’s ability to correctly identify unhealthy patients without misclassifying healthy ones. The F1-score adjusts Precision and Recall, providing a metric that balances the importance of avoiding FP and FN. Together, these metrics address the multifaceted challenge of medical diagnosis, ensuring the model’s performance is accurate, reliable, and clinically useful in distinguishing between healthy and disease classes.

## 4. Results

### 4.1. Performance of ML Classifiers

In this section, we demonstrate the performance of the ML classifiers (DT and RF) in distinguishing between healthy and unhealthy ERG signals. Several classification metrics were used for the evaluation to obtain a detailed understanding of the classifiers’ behaviors, including Accuracy, F1-score, AUC, Precision, and Recall. Each window has 21 combinations of size and overlap; thus, the mean and standard deviation of the metrics were taken for each window. In addition, the feature importance distributions for the top classifiers have been included. These distributions reveal the features that were given priority and deemed significant during training. They were instrumental in classifying the healthy and unhealthy signals.

Figure 4 shows the mean accuracies of the DT and RF ML models using various windows. Detailed values of the results are given in Appendix A. All the windows have the same upper bounds, with an Accuracy of 70.83% and AUC of 72.22% for RF and an Accuracy of 66.67% and AUC of 67.14% for DT. Table A1, Table A2, Table A3 show this in detail. The results showed minimal variation between the windows; some even had identical mean scores and standard deviations. It is worth noting that the RF-based classifiers still performed slightly better than their DT-based counterparts, as reported by the higher mean scores.

The feature importance distributions in Figure 5 indicate that the most important features are the maximum, average, and median intensities. The significance of these features alternates between the maximum and mean intensities, while the minimum intensity had little to no significance. This can be seen in the feature importance distribution of the other top classifiers in Appendix B. Figure A1 shows that bmean holds the highest significance for the model, with its median value approaching 0.4. This observation was consistent across all cases presented in Appendix B as can be seen in Figure A3 and Figure A4, except for Figure A2, where bmax exhibited a significantly larger interquartile range despite having a higher median value than bmean. In the remaining cases, the ranking of parameters by significance followed the sequence of bmax, bmean, bmedian, and bmin. Additionally, it is noteworthy that the variability in bmin was consistently the lowest across the figures, reinforcing its lesser impact on the model’s performance.

Figure 1 illustrates that both healthy and unhealthy signal spectrograms have low energy distribution within a similar range, whereas the medium and maximum energy distribution have a more comprehensive range and higher variation in intensities. As a result, the average intensities also exhibited a more pronounced difference between the healthy and unhealthy signals.

### 4.2. Performance of DL Classifiers

Figure 6 illustrates the results obtained from the different DL models (DenseNet, ResNet, VGG16, VGG19, and ViT) used in the analysis with various windows. These results showed higher variation between the windows, with the highest performance in terms of metrics being the Hamming window (Accuracy of 81.2% with ViT Small), regardless of the architecture. This window is well known for its ability to suppress the side lobes as sharply as possible while keeping the main one narrow enough [44]. The mentioned property allows one to extract most features from each window position. Among the architectures, ViT Small showed the best results. Figure 7a shows the receiver operating characteristic (ROC) curves for the ViT model with different windows. Figure 7b shows the ROC curves of different DL models with the Hamming window. Detailed results are provided in Table A4.

## 5. Discussion

STFT is a method based on the FT proposed as a solution to the lack of temporal information from the classical FT method. The use of windows that slide across the signal helps extract the spectral and temporal information of the signal and present it in the form of a spectrogram. However, since it is impossible to obtain high temporal and spectral resolution simultaneously, it is necessary to determine which window, window size, and overlap between the sliding window would yield the most optimal features of the ERG signal for classification. As shown in Appendix A, there was very little difference in the results for the window functions studied. However, we can see significant differences in performance depending on window size and overlap.

The classifiers with the best results were those with larger window sizes; given that larger window sizes provided better frequency resolutions, this could indicate that signals with higher frequency resolutions produce the most optimal features.

We can also observe that the RF-based models outperformed the DT-based models. This was expected given that RF uses an ensemble of DTs, hence being able to capitalize on multiple trees rather than a single individual tree to make its predictions. Table A3 shows that Boxcar and Bartlett have the highest mean scores and the most significant variance, AUC 69.3% and 64.4%, due to these windows having multiple high-value score classifiers. As reported in Table A1 and Table A2, it can be seen that all windows have nearly identical scores, with Boxcar and Bartlett having better classification Accuracy with higher scores than the others at 70.8%, suggesting that these windows might have slightly better effects on the extracted features than the rest. Thus, the models’ results and performance regarding the window functions were similar. However, there were differences in the metrics regarding the window size and overlaps. One plausible explanation for this was that the window function itself did not affect the signal significantly as much as the size and overlap of the windows do because the latter two determine the signal’s resolution. This effect is described in Appendix A where all windows have the same maximum value for each metric; however, the Boxcar window, which is a square window and does not change anything in the signal, has the highest mean and variance because it has multiple window sizes with the maximum value metrics. On the other hand, the Bartlett and Boxcar window functions have the best performance among the analyzed window types. The Bartlett window, with an almost triangular shape, is known for being used to prevent the generation of too many oscillations in the frequency domain [28]. The results of using the window are the same as for the basic Boxcar window. This was likely to be due to the relatively small and straightforward (interpreted) feature space. The STD analysis in Table A3 also shows that the smallest values are obtained for the Hamming and Hann window function cases for the RF decision algorithm and the Hann and Tukey window function for the DT algorithm.

This is different from DL methods. We assume that it is about automatic feature extraction. Figure 6 shows that the difference between the windowed features is more significant than the manual feature extraction approach. DL methods extract more features from the signal than can be extracted manually. Due to this, the average metrics values are higher than for the manual feature extraction cases. Comparing both approaches, we can conclude the perceptiveness of the modern DL approaches. However, we must also note that manual feature extraction with STFT can be considered the most explainable approach. Given that DL architectures do a better job at learning and extracting features at a wider scale, they can still be used as feature extractors alongside a classical model for the final classification. This will be explored in future research as it provides the potential to expand the feature space without the need for manual feature extraction.

## 6. Conclusions

This study investigated various window functions for STFT calculation (and spectrogram generation) to classify ERG signals. The spectrogram images were extracted using several combinations of well-known window functions, window sizes, and window overlap values, and the manual features were extracted to train the classical ML model using the same methods. Based on the comparison of the results of the two approaches, DL can be recommended. In terms of Accuracy, the ViT Small architecture with the Hamming window showed the best performance among the combinations of DL models with window types (81%). However, if manual feature extraction is required, a RF with a rectangular Boxcar window or Bartlett window can be recommended as an alternative to the DL approach. In the study, the mean Accuracy in these cases was 67.5%.

The results of the analysis of ERG using Short-Time Fourier Transform and ML techniques are, of course, dependent on the size of the dataset used for training, thus necessitating a large original sample. To address this limitation, expanding dataset volumes and promoting open data sharing within the electrophysiology community could enhance the diversity and representation of synthetic waveforms. Although these preliminary results have been generated with a relatively small sample set, it is one of the largest in the world by data quantity [38]. Moreover, we are actively developing larger synthetic datasets to support clinical studies [45].

Another limitation of this study was the feature space used in the ML approach; the analysis only used four features: the minimum, maximum, median, and mean brightness of the spectrogram; this could be a reason why there was little to no difference between the windows, this limitation was not encountered in the DL approach as the automatic feature extraction of DL architectures gave it access to a more prominent feature space. Hence, in future studies, we will look at expanding the feature space for the ML classifiers, as expanding the feature space for manual feature extraction approaches contributes to improved Accuracy and other metrics while maintaining the overall explainability of the system. This solution is essential for the developed algorithm to be easily understandable in medical applications.

## Figures and Tables

**Figure 1 bioengineering-11-00866-f001:**
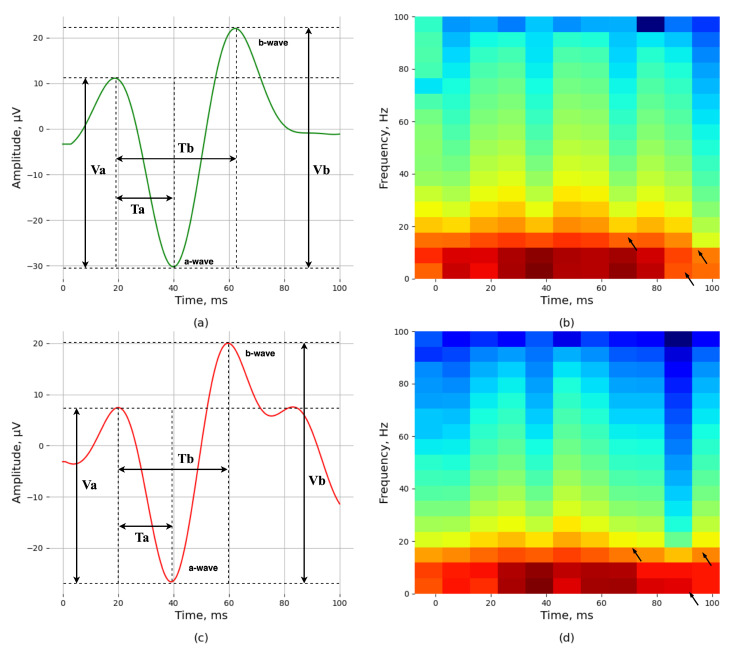
ERG representation of a healthy and an unhealthy Maximum (DA) 2.0 signal in time and time–frequency domains: (**a**) healthy signal in the time domain; (**b**) spectrogram representation of (**a**) in the time–frequency domain; (**c**) unhealthy signal in the time domain; (**d**) spectrogram representation of (**c**) in the time–frequency domain. The black arrows in (**b**,**d**) indicate that the spectrogram frequency distribution ranges from 0 to 80 ms and 0 to 15 Hz (maximum energy), 0 to 80 ms and 15 to 25 Hz (medium energy), and from 25 Hz (low energy). The key difference between the healthy signal (**b**) and the unhealthy signal (**d**) is evident in the energy distribution across the frequency bands. The healthy signal demonstrates a more diverse spread of energy, with maximum energy occurring at higher frequencies (0–20 Hz) compared to the unhealthy signal, where the maximum energy is concentrated at lower frequencies (0–15 Hz). This indicates that unhealthy signals tend to have more energy concentrated in the lower frequency bands, suggesting a potential marker for identifying signal health.

**Figure 2 bioengineering-11-00866-f002:**
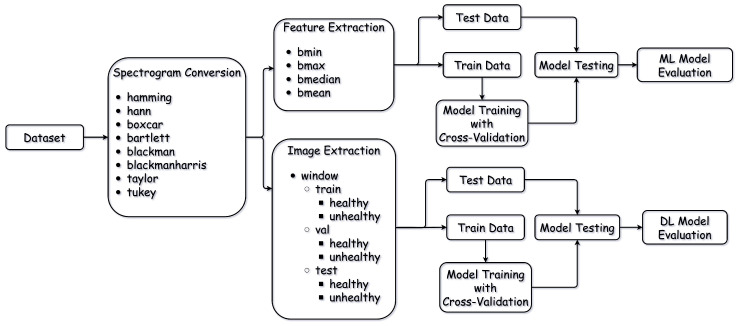
Complete study pipeline: from spectrogram conversion, feature extraction for ML, and image extraction for DL to data splitting and classifier evaluation.

**Figure 3 bioengineering-11-00866-f003:**
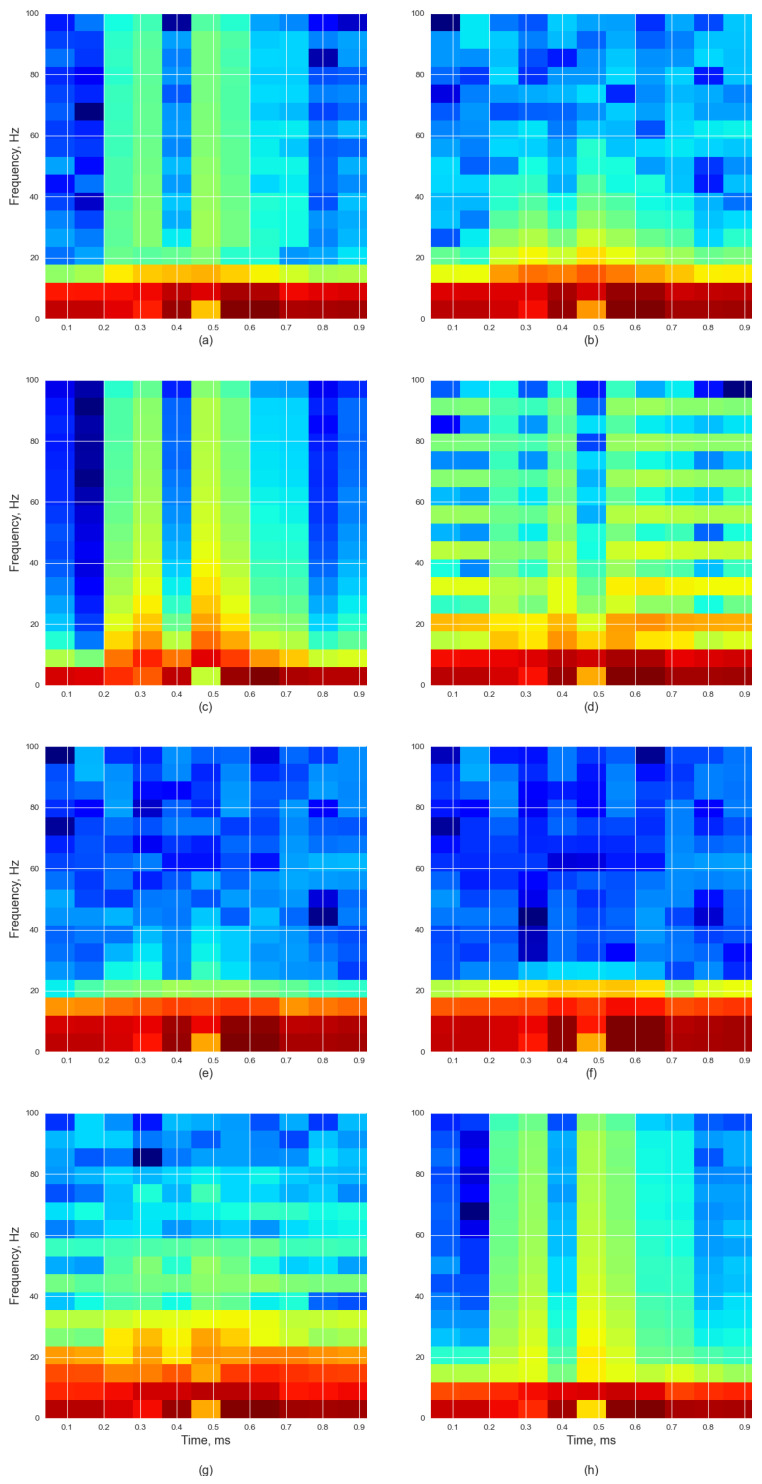
Spectrogram representation of signals based on windows used for the study with window size of 32 and an overlap of 16: (**a**) Hamming, (**b**) Hann, (**c**) Boxcar, (**d**) Bartlett, (**e**) Blackman, (**f**) Blackman–Harris, (**g**) Tukey, (**h**) Taylor.

**Figure 4 bioengineering-11-00866-f004:**
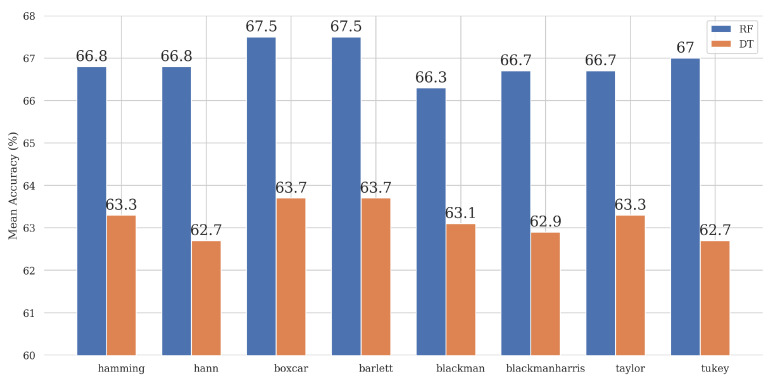
Mean accuracies of the analyzed DT and RF ML classifiers for all test windows.

**Figure 5 bioengineering-11-00866-f005:**
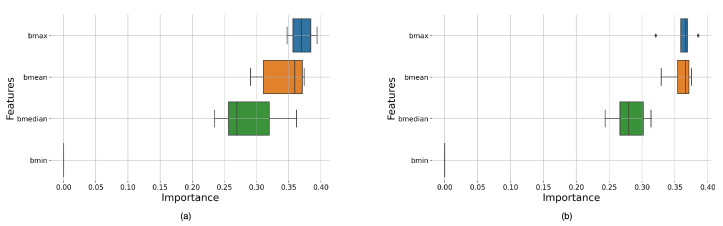
Feature importance distribution for classifiers with best performance. (**a**) RF classifier with Boxcar window, size 128, and an overlap of 64, (**b**) RF classifier with Bartlett window, size 128, and an overlap of 64.

**Figure 6 bioengineering-11-00866-f006:**
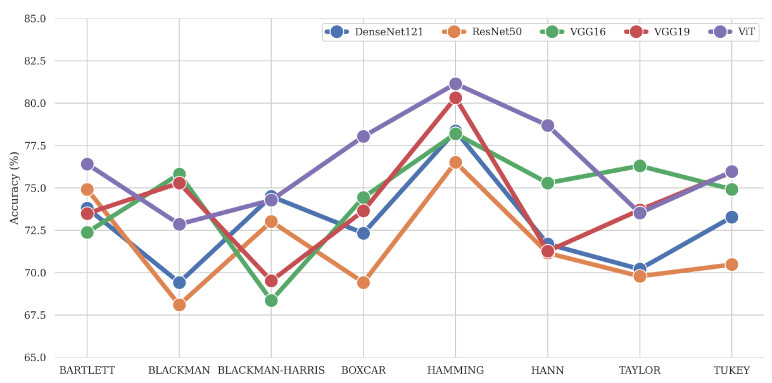
Classification accuracies of the analyzed DL architectures for all test windows.

**Figure 7 bioengineering-11-00866-f007:**
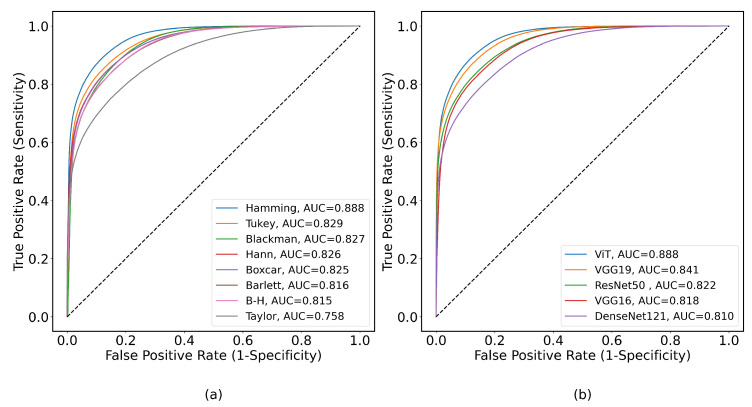
ROC curves for binary classification of ERG signals with corresponding AUCs for all tested windows using ViT model in (**a**) and for all tested models with Hamming window in (**b**).

**Table 1 bioengineering-11-00866-t001:** Related works.

Author(s)	Year	Method(s)	Signal(s)	No. of Subjects (Signals)
Gur et al. [13]	1979	FFT and LP	Corneal and Non-Corneal ERG	4 (N/A)
Gur et al. [7]	1980	FFT and LP	Corneal ERG	13 (N/A)
Van Der Torren et al. [14]	1988	FFT	Oscillatory Potentials	N/A (N/A)
Li et al. [15]	1990	Fourier Spectrum	Photopic and Scotopic ERGs	13 (23)
Sieving et al. [16]	1998	Discrete Fourier Transform	Flicker ERG	N/A (N/A)
Karimi et al. [17]	2012	Welch’s Power Spectral Density	Full-Field Photopic and Scotopic ERGs	N/A (54)
Alaql n.d. [19]	2016	Fourier Transform, STFT, CWT and DWT	Photopic ERG	N/A (N/A)
Zueva et al. [18]	2019	Fourier Series	Flicker and Pattern ERG	12 (N/A)
Erkyamaz et al. [20]	2021	STFT, CWT, DWT	Cone, Rod, Maximal ERG	40 (N/A)
Behbahani et al. [21]	2021	STFT, DWT	Photopic and Flicker ERG	20 (N/A)
Zhdanov et al. [8]	2022	CWT	Scotopic, Photopic, Maximum ERGs, Flicker and Oscillatory Potentials	N/A (425)
Kulyabin et al. [22]	2023	CWT	Photopic, Scotopic and Maximum	N/A (353)

**Table 2 bioengineering-11-00866-t002:** Distribution of healthy and unhealthy ERG signals, including balanced and unbalanced databases.

Unbalanced Dataset		Balanced Dataset	
**Healthy**	**Unhealthy**	**Healthy**	**Unhealthy**
60	143	60	62

**Table 3 bioengineering-11-00866-t003:** Table of window sizes and overlaps used for the spectrogram conversion. Each window was combined with each overlap for each iteration.

Window Sizes	Overlaps
128	64
64	32
32	16
16	8
8	4

**Table 4 bioengineering-11-00866-t004:** Table of hyperparameters used.

Hyperparameters	DT	RF
Criterion	Gini	Gini
Max depth	10	10
No. of estimators	N/A	250
OOB score	N/A	True
Min samples split	2	2
Min samples leaf	1	1

## Data Availability

Zhdanov, A.E.; Dolganov, A.Y.; Borisov, V.I.; Lucian, E.; Bao, X.; Kazaijkin, V.N.; Ponomarev, V.O.; Lizunov, A.V.; Ivliev, S.A. 355 OculusGraphy: Pediatric and Adults Electroretinograms Database, 2020. https://doi.org/10.21227/y0fh-5v04, accessed on 1 April 2024.

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
