# Peer review of "Electroretinogram Analysis Using a Short-Time Fourier Transform and Machine Learning Techniques"

_bioengineering, 2024, doi:10.3390/bioengineering11090866_

Round 1

Reviewer 1 Report

Comments and Suggestions for Authors

The manuscript addresses an important problem, incrementally advancing the knowledge in the investigated field; the work is methodologically correct, scientifically sound and technically interesting, and deserves to be published after minor revisions. 

Unfortunately, the overall quality of the manuscript is plagued by a large amount of writing mistakes, and a careful revision of the text is required.

1. The meaning of black arrows in Figure 1 (b) and (d) must be provided;

2. Figure 2 is unnecessary.

3. Cross-validation for deep learning models training is represented in Figure 3, but not stated in paragraph 3.2. The actual use of cross-validation needs to be clarified.

4. AUC and ACC values must be provided as percentage, with the proper symbol, or as decimal number.

5. The small sample size must be included in the limits of the study, especially for deep learning algorithms.

Author Response

Dear, reviwer. We thank you for your coooment. The answers you may find in the attched file. Also all changes are highlighted in the text of paper.

Reviewer 2 Report

Comments and Suggestions for Authors

General comment:

This work deals with the “optimized” classification of ERG signals based on STFT spectrograms and a ML decision system. The authors suggests different ML systems and windowing. 

Specific comments throughout the paper: 

1.

Minor edits:

Line 21, pg. 1:  missing space. Please proofread the manuscript for typos. 

Lines 31-37, pg. 2: to increase the paper quality and clarity, especially for the non-expert readers that may find your work (e.g., bachelor, master, phd students or some non-expert in the field) I suggest to refer immediately to Fig. 1 and explain the ERG signal as well as the wave components, amplitudes and time to peaks figures of merit. 

Lines 45-48, pg. 2: missing references. Please providing supporting refs. For deomonstrating that TD is the preferred approach. This is questionable. The literature gap should be better discussed.

Lines 48-51, pg. 2: unsupported statement. Please provide at least a ref.

The STFT formula in the introduction is something that is somehow not appropriate, since it is more a methodological part. The paper can be revised.

The novelty of the work is not stressed enough. 

3. 

The statistical information about the subjects (age, sex, etc.) involved in the datasets must be reported and commented for the sake of clarity. 

Line 190:195: this point is not clear at all. The signal reduction can have a large bias on the results. Did the authors test different lengths? A discussion is in order. The reliability of the results is questionable.

The feature extraction part must be better described. Lines 237-239 are not enough to ensure reproducibility. 

The language code used for the calculation is not reported. 

The ML part is quite standard, even though the dataset is quite narrow-sized. A discussion about this point is also in order. 

4. 

A statistical analysis of the extracted features should be presented. How do they vary across the dataset?

The authors did not perform an analysis aimed at identifying the most significant feature. Can the dimensionality of the problem be reduced?

5.

The discussion section is appreciated, but it lacks of a counterpart and an alternative source of information and comparison for the validation. 

6.

Conclusion section is fine. 

Comments on the Quality of English Language

The english language is somehow fine. Proofread the manuscript. 

Author Response

Dear Reviwer 2, we thank you for youe comments. The answes you may find in the attched file. As well, as all changes are highlighted in the paper text. 

Round 2

Reviewer 2 Report

Comments and Suggestions for Authors

I sincerely thanks the authors for the time spent in preparing the replies to all my questions and doubts and for implementing major modifications and improvements to the manuscript, that results in an enhanced quality and clarity. 

I do not have any further comments. 

Comments on the Quality of English Language

Minor editing of English language required